# SMC5/6-Mediated Transcriptional Regulation of Hepatitis B Virus and Its Therapeutic Potential

**DOI:** 10.3390/v16111667

**Published:** 2024-10-25

**Authors:** Johannes Bächer, Lena Allweiss, Maura Dandri

**Affiliations:** 1I. Department of Internal Medicine, Center for Internal Medicine, University Medical Center Hamburg-Eppendorf, Martinistr. 52, 20246 Hamburg, Germany; johannes.baecher@stud.uke.uni-hamburg.de (J.B.); l.allweiss@uke.de (L.A.); 2German Center for Infection Research (DZIF), Hamburg-Lübeck-Borstel-Riems Site, Germany

**Keywords:** hepatitis B virus, chronic hepatitis B, cccDNA, SMC5/6, NSE, transcription, HBx, antiviral therapy, siRNA, interferon

## Abstract

Cells have developed various mechanisms to counteract viral infections. In an evolutionary arms race, cells mobilize cellular restriction factors to fight off viruses, targeted by viral factors to facilitate their own replication. The hepatitis B virus (HBV) is a small dsDNA virus that causes acute and chronic infections of the liver. Its genome persists in the nuclei of infected hepatocytes as a covalently closed circular DNA (cccDNA) minichromosome, thus building up an episomal persistence reservoir. The chromosomal maintenance complex SMC5/6 acts as a restriction factor hindering cccDNA transcription, whereas the viral regulatory protein HBx targets SMC5/6 for proteasomal degradation, thus relieving transcriptional suppression of the HBV minichromosome. To date, no curative therapies are available for chronic HBV carriers. Knowledge of the factors regulating the cccDNA and the development of therapies involving silencing the minichromosome or specifically interfering with the HBx-SMC5/6 axis holds promise in achieving sustained viral control. Here, we summarize the current knowledge of the mechanism of SMC5/6-mediated HBV restriction. We also give an overview of SMC5/6 cellular functions and how this compares to the restriction of other DNA viruses. We further discuss the therapeutic potential of available and investigational drugs interfering with the HBx-SMC5/6 axis.

## 1. Introduction

Chronic infection with the hepatitis B virus (HBV) represents a major health burden with global impact. Despite the availability of an efficient prophylactic vaccine, an estimated 296 million people are chronically infected worldwide. These patients have a high risk of developing liver cirrhosis and hepatocellular carcinoma (HCC), leading to more than 800,000 deaths annually [1].

HBV is a small, enveloped virus with a relaxed, circular, partially double-stranded DNA genome (rcDNA), which has developed unique replication modalities involving the reverse transcription of an RNA intermediate, the pre-genomic RNA (pgRNA) [2]. Shortly after cell entry, which is mediated by binding to the hepatocyte-specific sodium-taurocholate co-transporting polypeptide (NTCP), the nucleocapsid disassembles at the nuclear pore complex and the rcDNA is released into the cell nucleus. As a “lesion-bearing” molecule, the rcDNA needs to be converted by the host cellular machinery into a covalently closed, circular DNA episome (cccDNA); a process that occurs concomitantly with the deposition of histone and non-histone proteins to form a stable minichromosome [3]. The chromatinized cccDNA serves as a template for the transcription of all viral genes. These include the 3.5 kb-long pgRNA, the precore mRNA (3.5 kb), the preS1 mRNA (2.4 kb), preS2/S mRNA (2.1 kb), and X mRNA (0.7 kb), as well as spliced RNAs of varying lengths. The cccDNA represents the central molecule for HBV persistence since its formation and maintenance are essential to produce the pgRNA, which is encapsidated together with the HBV polymerase and reverse transcribed into a new rcDNA molecule. Mature nucleocapsids are enveloped in host membranes studded with the HBV surface antigens (HBsAg) and secreted into the bloodstream. Remarkably, aberrant reverse transcription of the pgRNA can also lead to the formation of a double-stranded linear genome that serves as a substrate for the integration of HBV DNA sequences into the host chromosomal DNA [4]. Even though integration occurrence is not required in the HBV life cycle, the presence of integrated viral sequences can substantially contribute to the production of HBsAg.

The host factors involved in cccDNA formation and chromatinization [3,5,6], and those affecting cccDNA activity and persistence in infected cells, are not fully understood. Histones are susceptible to post-translational modifications (PTMs) that can activate or inactive gene transcription. Histone modifications on the cccDNA were shown to resemble those on the host chromatin, although the distribution of PTMs on the cccDNA appeared rather heterogeneous in chronic hepatitis B (CHB) patients [7,8]. Not only cellular factors but also viral proteins, such as the regulatory X protein (HBx), are recruited on the cccDNA [9].The critical role that HBx plays in HBV replication was highlighted by studies with HBV X-minus mutant viruses, which demonstrated that HBx is essential for cccDNA-driven viral transcription [10,11]. Moreover, HBx was shown to mediate the degradation of the structural maintenance of chromosomes 5/6 complex (SMC5/6) [12,13], a DNA-binding complex with multiple functions in DNA repair and the maintenance of genomic stability [14]. SMC5/6 can act as a host restriction factor in HBV infection by counteracting cccDNA activity (Figure 1A). Despite the increasing awareness of the role of SMC5/6, many open questions remain, including the impact of antiviral treatments on SMC5/6-mediated silencing of the HBV reservoir in infected human hepatocytes. The limited host range and high liver tropism of HBV have hindered research focusing on cccDNA metabolism, which requires a deep understanding of the interactome established in infected human hepatocytes and the innate and adaptive immune responses elicited by the infection.

## 2. The SMC5/6 Complex and Its Functions on Cellular Chromatin

As a member of the family of structural maintenance of chromosomes (SMC) complexes, the SMC5/6 complex maintains the structural integrity and spatial organization of chromosomes [15]. The family also comprises condensin and cohesin, the canonical and most extensively studied chromosomal maintenance complexes. These complexes all enable the spatial organization of the genome through the folding and looping of chromosomes throughout the different phases of the cell cycle. In addition to ensuring the integrity of the cellular genome, spatial chromatin organization represents an independent mechanism regulating chromatin accessibility during gene transcription and DNA repair. SMC complexes are ring-shaped protein complexes, comprising two SMC proteins, containing long coiled coils that dimerize through a “hinge domain” on one end and a “head domain” on the other. The head domain forms the ATPase. All three SMC complexes contain specific pairs of SMC proteins and associate with varying numbers of non-SMC elements (NSE). SMC1 and SMC3 form the core of the cohesin complex, which holds sister chromatids together after replication, thus ensuring error-free segregation of chromatids during mitosis. Cohesin also establishes topologically associating domains (TADs) during interphase, large chromatin segments with defined contact areas between chromosomes. Condensin contains SMC2 and SMC4 proteins at its core and is best known for its function in chromosome condensation during mitosis. On a broader view, SMC complexes shape the structure of chromosomes through an intrinsic loop extrusion activity, creating DNA loops of varying sizes and complexity, and eventually compacting and folding chromosomes in space [16]. While the two canonical complexes facilitate the global folding of large regions or entire chromosomes during defined cell cycle phases, SMC5/6 appears to act more locally to compact smaller chromosome regions. The mechanisms of folding and compaction involve ATP-dependent DNA binding and topological entrapment of the DNA. The compacted DNA is believed to isolate those structures from undesirable effects or create a protective environment for the recruitment and activation of downstream effector proteins. In this sense, SMC5/6 complexes have been described as “micro-compaction machines” [16] and “loop extrusion motors” [17].

The best-studied functions of SMC5/6 are the promotion of double-strand break repair via homologous recombination and the maintenance of replication fork stability. Hence, the complex plays a crucial role in facilitating proper DNA replication in the presence or absence of DNA damage. Replication creates topological stress by supercoiling of DNA, which, if not attended to, can stall replication. SMC5/6 was shown to stabilize sister-chromatid intertwines and release topological stress through the interaction with topoisomerase or the rotation of the replication fork. SMC5/6 functions are particularly necessary in genomic loci that are difficult to replicate such as the rDNA locus, heterochromatin, and telomeres (for a current review on SMC5/6 genomic functions, refer to [14]).

SMC5/6 complexes contain up to six additional NSE subunits (Nse1-6 in yeasts and NSE1-4 in mammals, also called non-structural maintenance of chromosomes element 1 through 4 (NSMCE1-4)), forming three distinct subcomplexes, i.e., Nse1/Nse3/Nse4, Nse2/SMC5, and Nse5/Nse6. In humans, SMC5/6 complex localization factors 1 and 2 (SLF1 and SLF2) have been implicated as functional homologs of the yeasts Nse5 and Nse6, respectively. The distinct properties of these NSE sub-complexes are believed to confer functional diversity and context specificity in maintaining genomic integrity. Indeed, a unique feature of these NSE elements is their ability to function as post-translational modification enzymes. NSE1 is a ubiquitin ligase, while NSE2 catalyzes the transfer of Small Ubiquitin-Related Modifier (SUMO) proteins, shown to interact with a growing number of substrates, including components of the complex itself. Another defining property of SMC5/6 is its binding affinity towards unconventional DNA structures such as branched DNA and ssDNA-dsDNA junctions. Intriguingly, both yeast and human SMC5/6 were shown to bind and stabilize supercoiled, i.e., over-twisted, DNA structures [16,18].

Notably, the restriction of episomal DNA is a function unique to SMC5/6. It is therefore not surprising that SMC5/6 has also been implicated in the inhibition of other dsDNA viruses, such as herpes viruses, papillomaviruses, polyomaviruses, and unintegrated human immunodeficiency viruses (HIV) [19]. As data on the interplay between SMC5/6 and those viruses are emerging, it appears that SMC5/6 acts as a general intrinsic antiviral defense mechanism involved in the silencing of invading extrachromosomal genomes. Whether this is accomplished using a unified recognition and effector mechanism for all viruses or whether selective activities within the SMC5/6 sub-complex are responsible for specific viral interactions is still unclear. Given the diverse structure and the cellular context of these viruses, the nature of these interactions will likely have been adapted to the particular invader. However, as all these viruses exist as extrachromosomal genomes, mainly in a circular form, restriction often involves genome compaction, epigenetic repression, SUMOylation, and localization to specific sub-nuclear structures. Likewise, many of these viruses have adopted mechanisms to antagonize the cell’s defense system by expressing viral proteins (such as the BNRF1 protein of the Epstein Barr virus [20]) able to degrade components of the SMC5/6 complex.

## 3. Mechanism of HBx-Mediated Degradation of SMC5/6

The small 17kDa HBx protein has been reported to interact with many host factors and to affect several cellular pathways. Still, its central function in promoting cccDNA-driven transcription has long remained enigmatic [21]. It has been known since the 1990s that HBx binds the DNA-damage binding protein 1 (DDB1) and that mutations preventing HBx interaction with DDB1 inhibit HBV infection [22]. HBx was also known to operate as a transactivator of episomal DNA, able to act both on HBV DNA and unrelated plasmids, but unable to activate the transcription of chromosomal genes [23]. DDB1 binds Cullin 4 and forms a cullin4-RING ubiquitin ligase complex. In this context, HBx was postulated to serve as a DDB1–cullin-4-associated factor (DCAF), with the primary function of making contact with substrate proteins to promote their ubiquitinylation and proteasomal degradation. In 2016, two groups independently identified the SMC5/6 complex as a target for the HBx-DDB1 E3 ligase complex [12,13]. Indeed, SMC5 and SMC6 levels were reduced in HBx-expressing and HBV-infected cells. Moreover, the SMC5/6 complex was shown to bind the cccDNA in the absence of HBx, and the loss of SMC5/6 function rescued the replication of an HBx-deficient virus. These data identified SMC5/6 as a host restriction factor that can recognize and transcriptionally silence cccDNA. SMC5/6 antagonism by HBx is evolutionarily conserved across hepadnaviruses in mammals [24] and is now considered the primary function of HBx in HBV infection.

In HBV-infected cultured hepatocytes, HBx is expressed early after infection and locates to the nucleus [25]. How HBx interacts with DDB1 and with SMC5/6 is still not entirely solved. HBx contains a total of 154 residues, and residues 88–100 form a conserved alpha-helical motif called the H-box. HBx mutagenesis studies have suggested that the H-box and some adjacent residues are necessary for DDB1 binding [26]. Recent biophysical and biochemical analyses mapped the minimal region required for DDB1 binding to residues 45-140 [27]. Which component of the SMC5/6 complex interacts with HBx is not fully clarified. Several interaction studies failed to identify a single subunit of the SMC5/6 complex as an HBx interacting partner, suggesting that more than one subunit is involved in a stable interaction [28,29]. In vitro ubiquitylation assays showed that SMC5 and SMC6 were directly polyubiquitinated by the Cullin4 E3 ligase upon binding to HBx [12]. Since depletion of almost any SMC5/6 subunit results in complex disruption through the degradation of the other components, polyubiquitination of a single subunit seems sufficient for inhibiting viral restriction [13,29]. Interestingly, HBx was shown to degrade SMC5/6 independently of its binding to the cccDNA or its sub-nuclear localization [29]. Dissecting the structure of HBx remains challenging, and more functional and structural research is needed to clarify the interaction of HBx with DDB1 and SMC5/6. This knowledge could potentially be harnessed to develop antiviral treatments, since targeting HBx by interfering with the DDB1-HBx-SMC5/6 axis would prevent SMC5/6 degradation and induce cccDNA silencing. Likewise, the mechanism of SMC5/6-mediated silencing of cccDNA deserves further investigation as this process could also be exploited for therapeutic approaches.

## 4. Mechanism of SMC5/6-Mediated Silencing of cccDNA

The mechanism and sequence of events involved in SMC5/6 recognition of HBV cccDNA are not fully understood (Figure 1B). The first step in the silencing process consists of recognizing the cccDNA and, hence, must involve a mechanism to distinguish between chromosomal and extrachromosomal DNA. Yeast Smc5/6 was shown to recognize transcription-induced positively supercoiled DNA [30]. Upon binding, Smc5/6 dimers initiate loop extrusion to fold the supercoiled DNA into large plectonemic loops for the three-dimensional organization of chromosomes in the interphase. This is thought to ensure transcription management on a genome-wide scale by insulating or resolving supercoiled areas. In a recent study, the human SMC5/6 complex was shown to localize to a few highly transcribed and positively supercoiled chromosomal loci [31]. Moreover, SMC5/6 bound episomal DNA independently of its origin or chromatin composition but failed to associate with linear extrachromosomal DNA. This highlights an unforeseen role of SMC5/6 in topology management in interphase but also suggests that sensing DNA topology, rather than a specific sequence or chromatin composition, detects and counteracts invading DNA molecules.

The cccDNA is known to be a supercoiled molecule, and its transcription is expected to build up positive supercoils, which may promptly recruit the SMC5/6 complex and promote the entrapment of the viral minichromosome. While cccDNA repair processes that are necessary to transform the rcDNA genome into cccDNA and its chromatinization appear to occur independently of HBx [5], early cccDNA transcription and production of HBx in newly infected cells appear necessary to prevent SMC5/6-mediated cccDNA silencing. This is in line with cell culture data showing that HBx is the first protein to be expressed [25] and that a fraction of infected cells harbors cccDNA molecules transcribing only HBx transcripts, which might resemble early infection events [32]. Of note, other studies suggested that HBX mRNA, which has been detected in the plasma of infected individuals, may be present within HBV infectious particles and translated before de novo HBx RNA transcription starts. According to this scenario, it is conceivable that such HBX transcripts could also contribute to cccDNA activation and counteract SMC5/6-mediated suppression of the HBV reservoir [33,34,35].

Functional complementation assays using mutants of the individual subunits of SMC5/6 were applied to define the subunits and enzymatic functions responsible for binding episomal reporter DNA constructs [29]. SMC5 and SMC6 mutants defective in ATP binding or ATP hydrolysis failed to bind to episomal DNA, suggesting that ATPase-dependent topological entrapment of the DNA is a prerequisite for silencing. NSE4A, as part of the NSE1/NSE3/NSE4 subcomplex, was also found to be essential for binding episomal DNA. Given that SMC5/6 has several DNA binding sites within the complex and undergoes different conformational changes associated with DNA binding to support functional changes in the DNA configuration, further studies need to investigate the precise nature of DNA binding and the role of its NSE partners. Since loop extrusion and topological binding are thought to represent two distinct functional states of SMC5/6 [17], it will be interesting to determine if the cccDNA is merely entrapped or further compacted through loop extrusion.

Within the nucleus, cccDNA silencing takes place inside subnuclear structures called nuclear domains 10 (ND10) or promyelocytic leukemia (PML) nuclear bodies. ND10 are large dynamic protein condensates, which contain PML and numerous other proteins, including Sp100, DAXX, or SUMO [36,37]. ND10 plays a role in many cellular homeostatic pathways, such as transcription, DNA damage response, and host antiviral defense. PML expression is strongly induced in response to interferons (IFN), and both the size and number of ND10 bodies increase upon IFN stimulation. ND10 components traffic to the incoming genomes of many DNA viruses and restrict transcription from these extrachromosomal viral templates, while other viruses usurp ND10 for efficient viral propagation. Many DNA viruses encode regulatory proteins that mediate the degradation or disruption of ND10 components, such as herpes viruses. The study by Niu et al. showed that SMC5/6 co-localizes with ND10 in uninfected human hepatocytes [33]. However, HBV infection did not alter the expression of PML and Sp100 despite HBx-mediated degradation of SMC5/6 from ND10. Yet, the depletion of both PML and Sp100 dispersed SMC6 from nuclear bodies, resulting in the loss of episomal restriction activity. Consequently, HBV transcription could take place in the absence of HBx. Together, these data demonstrate that SMC5/6 suppresses HBV transcription only when localized to ND10.

By performing RNA-Seq in cultured primary human hepatocytes (PHH), Niu and colleagues did not detect prominent changes in the host transcriptome, neither shortly after HBV infection when SMC5/6 was still present, nor at later times post-infection when SMC5/6 was degraded. Both HBV and an HBx-negative virus could establish infection without inducing the expression of ISGs or the production of other cytokines [33]. These observations are in line with previous studies in chimpanzees and patients, which show a lack of substantial intrahepatic innate immune response upon HBV infection [38,39]. These data therefore suggest that SMC5/6 acts as an intrinsic antiviral restriction complex that recognizes supercoiled cccDNA minichromosomes and turns off HBV transcription without triggering an innate immune response. Its antiviral function is likely mechanistically related to its role in cellular genome maintenance and does not rely on the help of classical innate effector proteins.

SLF1 and SLF2 have been described as mammalian orthologs of Nse5 and Nse6, forming a stable sub-complex binding to the head domains of SMC5/6, which are involved in localization to cellular chromatin [40]. Abdul and colleagues showed that SLF2, but not SLF1, is required for the restriction process at a step downstream of DNA binding. Precisely, SLF2 appears responsible for the recruitment of SMC5/6-bound episomes to ND10 nuclear bodies. Using an alternative approach, based on a small interfering RNA (siRNA) screen targeting 91 ND10-related proteins, Yao and colleagues recently identified SLF2 as a necessary factor for the recruitment of cccDNA to ND10 [41]. Importantly, these assays were performed in HBV-infected HepG2-NTCP cells using wildtype or an HBx-deficient mutant and thus corroborated the importance of SLF2 in the context of the HBV cccDNA. This study also used FISH assays in PHH to directly visualize cccDNA from transcriptionally inactive HBx-deficient mutants in ND10 bodies. In the case of HIV infection, all components of the SMC5/6 complex, including SLF1 and SLF2, appeared to be required to compact and achieve effective silencing of unintegrated HIV-1 DNA [42,43]. Another study showed the ability of a sub-complex consisting of SUMO Interacting Motifs Containing 1 (SIMC1) and SLF2 to direct SMC5/6 to polyomavirus replication centers at ND10 nuclear bodies [44]. Although the SIMC1/SLF2 sub-complex was shown to recruit SMC5/6 to host DNA lesions, the SIMC1/SLF2-based interaction, via their SUMO-interacting motifs in SIMC1, could constitute a general mechanism to recruit SMC5/6-bound viruses to SUMO-rich ND10 bodies. It will be interesting to determine whether SMC5/6 and its subcomplex SIMC1/SLF2 play a role in the life cycle of other viruses known to associate with ND10. 

After DNA binding and recruitment to ND10, the last step in this silencing cascade is the restriction of gene expression from the episomal template. Since SMC5/6 binding requirements are similar for chromosomal and extrachromosomal DNA, i.e., topological recognition of supercoiled DNA [31], the restriction activity of the complex likely relies on subsequent steps. To date, little is known about the actual restriction process. After the initial binding at the plectoneme, SMC5/6 could either topologically entrap the cccDNA to hinder its transcription or initiate loop extrusion to compact the molecule further. Interestingly, the SMC5/6-mediated restriction of both Kaposi’s sarcoma-associated herpesvirus (KSHV) [45] and unintegrated HIV DNA [42] was accompanied by further compaction of the viral DNA. 

As the subcomponents of SMC5/6 are recognized to facilitate and regulate specific functions of the entire complex, deciphering the roles of its individual components in viral restriction remains challenging. The SUMO E3 ligase NSE2 interacts with the coiled-coil arm domains of SMC5. Complementation assays with functionally mutated NSE2 revealed that this protein was not required for SMC5/6 binding to episomal DNA or for its colocalization to ND10 and suggested its involvement in the restriction process downstream within the context of ND10 [29]. The SUMO E3 ligase activity of NSE2, however, was not required for its repressive function. This is a surprising finding given the important role of SUMOylation in the cellular functions of SMC5/6 and to effectively silence unintegrated HIV-1 genomes [43]. The function of NSE2 and other complex members in the restriction process of HBV cccDNA thus awaits further clarification.

The mechanism of SMC5/6-mediated silencing may also lead to a rearrangement of the epigenetic landscape of the cccDNA. Repressive epigenetic modifications are found on the cccDNA in the absence of HBx [9,46]. Using HBx-deficient virus, silenced cccDNA was found to be associated with reduced acetylation levels of H3 and H4 histones, and increased levels of H3K9me3 and HP1 [46], together with the recruitment of histone deacetylases, including HDAC1 [9] and histone methyltransferase SETDB1 [46,47]. Thus, the cccDNA epigenetic landscape described in the absence of HBx resembles that of transcriptionally silent heterochromatin. It is worth noting that although DDB1 is the best-characterized binding partner of HBx, additional HBx-binding partners have been described. These include B-cell lymphoma (BCL)-2 family proteins, like Bcl-xL, which have been shown to promote viral replication in vitro and in vivo [48,49,50], and Spindlin 1, an epigenetic reader that was recently reported to be hijacked by HBx to enhance cccDNA transcriptional activity [51]. Moreover, the host complex HMGB1 was identified as an additional host factor antagonized by HBx and involved in cccDNA silencing, since both SMC5/6 and HMGB1 were found to be associated with the cccDNA in the absence of HBx [52]. However, little is known about how SMC5/6 may interact with classical epigenetic modifiers to affect cellular or episomal chromatin. It is plausible that the cccDNA undergoes epigenetic reprogramming in ND10, which is known to be rich in histone chaperones and chromatin modifiers including HP1 [53], and is then shuttled to heterochromatin regions in the nucleus [54]. More research is needed to investigate SMC5/6-dependent epigenetic changes and to understand how epigenetic changes contribute to the repression of cccDNA transcription and maintenance. 

Taken together, SMC5/6 appears to recognize and bind cccDNA because of its supercoiled configuration after initial transcription (Figure 1B). The binding depends on SMC5 and SMC6´s ATPase function and NSE4A. Subsequent recruitment to ND10 via SLF2 then facilitates transcriptional restriction. The restriction process might involve topological entrapment or further compaction through loop extrusion and epigenetic repression through the interaction with NSE2 and other cellular factors in the environment of ND10 nuclear bodies. It needs to be mentioned that many studies on the cellular functions of SMC5/6 were performed in yeast since the regulation of this complex in human and yeast cells is highly conserved. Moreover, studying the HBx-DDB1-SMC5/6 interplay during natural infection events with wild-type HBV remains challenging due to the limited availability of human liver biopsies and specific antibodies to study HBx and its interacting host proteins in infection systems. Nevertheless, SMC5/6 was shown to be degraded in cultured human hepatocytes [12,13,25,33] and human liver chimeric mice infected with HBV [13,55].

## 5. Harnessing the HBx-SMC5/6 Axis for Antiviral Therapy

The ideal endpoint of CHB treatment is a sterilizing cure, which would require the elimination of the viral minichromosome and integrated HBV DNA sequences. While achieving this goal does not seem attainable, current treatments in development aim for a functional cure, defined as the sustained loss of HBsAg and HBV DNA suppression, with or without the appearance of anti-HBs antibodies. Clearance of HBsAg is associated with a lower risk of relapse after treatment discontinuation and HCC development compared with only HBV DNA suppression. 

Two classes of therapies are currently approved for CHB: nucleos(t)ide analogs (NAs) and pegylated interferon alpha (pegIFNα). NAs efficiently suppress viral replication by inhibiting the HBV reverse transcriptase and substantially improve liver pathology. However, they neither target cccDNA directly nor limit its transcriptional activity. Consequently, HBV transcripts and proteins continue to be produced, and NA-based treatment rarely achieves a sustained off-treatment control of HBV replication, generally requiring many years to lifelong treatment [56]. Conversely, IFNα can act as an immunomodulatory agent and induce direct antiviral effects such as repressing cccDNA activity and promoting its destabilization [57,58]. IFNα has been shown to be superior to NA therapy because higher rates of HBsAg loss can be achieved. However, a finite therapy (1 year) with pegIFNα is associated with substantial side effects, and functional cure rates remain limited. Therefore, developing new antivirals and combination therapies with a finite duration is needed to achieve higher rates of functional cure. Given the key roles that cccDNA plays in HBV persistence and HBx in maintaining cccDNA activity, therapeutic strategies promoting the destabilization/reduction of the HBV reservoir in the liver and its functional silencing appear essential to achieve the sustained control of HBV infection. 

Multiple host and viral factors have been shown to regulate cccDNA activity at the epigenetic level [47]. Epigenetic changes and substantial differences in cccDNA activity have been reported among different phases of CHB [59] and also following several years of NA treatment [60]. Yet, interventions targeting the HBx-SMC5/6 axis may induce cccDNA silencing and the sustained control of HBV infection (Figure 2). It must be noted that prolonged degradation or impairment of the cellular SMC5/6 complex, as is expected to occur in HBV-infected hepatocytes in the liver of CHB individuals, has been associated with the accumulation of spontaneously induced DNA damage, p53 activation, and even mitotic aberrations [61,62]. Thus, these studies highlight the importance of developing therapeutic strategies targeting HBx production and/or function, since HBx abrogation would restore SMC5/6 function in liver cells and likely contribute to reducing liver disease progression and HBV-related carcinogenesis.

### 5.1. Direct Targeting of HBx Function

Targeting the HBx protein directly remains challenging as the 3D structure of the full-length protein still needs to be solved [63,64]. Using antibodies in a therapeutic context is also a difficult task since HBx is not expressed on the cell surface and predominantly resides inside the nucleus [25]. Nevertheless, in the past years, various substances have been identified and assessed for their ability to target HBx and its function. Cheng et al. identified dicoumarol, a natural anticoagulant, to affect intracellular HBx levels [65]. Besides its antithrombotic functions, dicoumarol competes with NAD(P)H for NAD(P)H:quinone oxidoreductase (NQO1) binding and, therefore, antagonizes its function as a cytosolic reductase, which was also found to regulate 20s proteasomal degradation of host proteins, including p53 [65]. Co-IP studies showed that NQO1 binds to HBx and prevents HBx degradation in a ubiquitin-independent process, but most likely via the 20s proteasome. The study also confirmed the increase in SMC5/6 upon NQO1 knock-out in HBV-infected cells and the decrease in cccDNA transcription. Notably, the study showed that repression of the episomal HBV DNA could also be achieved in the setting of established infections both in vitro and in vivo and that the transcriptional status correlated with corresponding chromatin marks H3K9ac and H3K4me3 (active) and H3K9me3 and H3K27me3 (repressive) in HepG2-NTCP cells and a recombinant cccDNA mouse model, in a pattern resembling HBx deficiency [65]. Nevertheless, interactions between dicoumarol and several host pathways, including vitamin K antagonism, raise substantial safety concerns. A large screening of small molecules also identified the macrolide compound rapamycin [66], which is used in the clinic to prevent acute rejection after organ transplantation, for its ability to reduce the stability of the HBx protein in HBV-infected cells by promoting its degradation via the ubiquitin–proteasome system [65,66]. The rapamycin-mediated degradation of HBx led to the inhibition of cccDNA transcription, which could be demonstrated in vitro and in vivo using a recombinant cccDNA mouse model. 

Sekiba and colleagues [67] also developed an elegant high-throughput screening assay and demonstrated the ability of nitazoxanide (NTZ), a thiazolide anti-infective compound that is approved by the FDA for treating viral and protozoan enteritis, to hinder the interaction of HBx with DDB1. The study could show NTZ-mediated restoration of SMC5/6 and suppression of HBV transcription in HBV-infected human hepatocytes. Nevertheless, the NZT activity in suppressing HBV replication appeared moderate, and further optimization of the compound may be necessary to develop into a clinically efficacious anti-HBV drug. The ubiquitination of SMC5/6 requires an additional ubiquitin-like protein for activation, the so-called neuronal precursor cell-expressed developmentally down-regulated protein 8 (NEDD8). It was shown that pevonedistat, a NEDD8-activating enzyme inhibitor, could restore SMC5/6 protein levels, hence suppressing HBV transcription in in vitro HBV replication models and human hepatocytes [68]. Using a different approach, another study demonstrated the ability of a cell-penetrating monoclonal antibody targeting HBx, which was developed by conjugating the antibody with the cell-penetrating peptide of HIV Trans-Activator of Transcription (Tat)-1 protein, to suppress viral replication and protein production in cells and mouse models [69]. Although significant decreases in HBsAg and HBV DNA levels were observed in mice after a single infusion, virological rebound occurred rapidly, indicating that repeated administration improved cell delivery and efficacy are needed.

### 5.2. RNAi-Based Treatment to Induce SMC5/6-Mediated Suppression of HBV Transcription

The difficulties that may be encountered by developing safe and highly effective small molecules targeting directly the regulatory HBx protein can be overcome by interfering with HBx on the RNA level. This can be either accomplished by RNA interference (RNAi) with small interfering RNAs (siRNA) or antisense oligonucleotides (ASOs). The main aim of RNAi-based treatments is to lower HBV replication and the levels of circulating viral antigens, such as HBsAg, in treated patients [70] and eventually achieve a functional cure. Small interfering RNAs are short, double-stranded RNA sequences where the antisense strand is complementary to the target sequence of mRNA and guides the molecule to the RNA-induced silencing complex (RISC). The RISC complex cleaves the targeted mRNA promoting its degradation [71,72]. In contrast, ASOs are short single-stranded DNA molecules that complementarily bind the targeted mRNA transcripts. The resulting DNA–RNA hybrid recruits RNase H for site-specific cleavage [71,72]. Several RNAi therapeutics for HBV are in various stages of preclinical and clinical development and have demonstrated varying abilities to reduce the levels of circulating HBsAg, with some candidates showing sustained HBsAg loss after cessation of therapy [73]. The addition of chemical modifications increases the half-life of RNAi compounds and enhances their efficacy at lower doses, thus reducing their administration frequency. 

Moreover, the systems that deliver siRNAs into target cells represent crucial elements in designing effective treatment strategies [70,73]. Both lipid nanoparticles (LNP) (i.e., ARB-1467) and, more frequently, the conjugation of siRNAs to N-acetylgalactosamine (GalNAc/NAc) (ALG-125755; RG-6346; AB729; VIR-2218; JNJ-3989) have been used for liver-targeting therapy. Derivatives of GalNAc, with enhanced binding to asialoglycoprotein receptors on hepatocytes, enable the enhanced delivery of siRNAs to the hepatocytes. To be noted, ASOs can freely enter the cells, and studies in murine models showed that unconjugated ASOs are taken up predominantly by non-parenchymal cells in the liver. In contrast, most Gal-NAc-conjugated ASOs (>70%) concentrate in parenchymal cells [74]. Among the ASOs designed and investigated in CHB, those with Gal-NAc conjugates (RO7062931, Roche and ALG-020572, Aligos) have shown a moderate reduction of HBsAg in patients. The most promising candidate currently in development for the treatment of CHB is Bepirovirsen (GSK3228836), a subcutaneously administered 2′-O-methoxyethyl unconjugated antisense nucleotide. Phase 2 studies showed that Bepirovirsen resulted in sustained HBsAg and HBV DNA loss in 9 to 10% of patients for week 24 after treatment [75]. The reasons for the superior efficacy of this ASO are not clear but are likely due to its ability to stimulate immune reactions via pattern recognition receptors, such as TLR8, in non-parenchymal cells.

Due to the unique genomic organization of the HBV genome, all HBV mRNAs transcribed from the cccDNA share a common polyadenylation signal located in the core ORF region [76]. Thus, HBV RNAi targeting the X ORF region, which overlaps with all other HBV ORFs, would not only hinder the translation of pgRNA and subgenomic RNAs responsible for the production of virions and HBsAg, but would also interfere with the short (0.8kb) HBx RNA. By abrogating HBx production and HBx-mediated degradation of the SMC5/6 complex, transcriptional suppression of cccDNA will occur during treatment. The silencing of the cccDNA minichromosome may therefore represent a secondary and fundamental mode of action of siRNA-based therapies. 

To study the ability of siRNAs targeting all HBV RNAs to also lower HBx production in infected human hepatocytes in vivo, LNP-formulated siRNA was injected for a period of 4 weeks in HBV-infected human liver chimeric mice and virological changes were assessed in the serum and liver [55]. This study clearly demonstrated a significant reduction of serological and intrahepatic viral markers, including the decrease in HBx protein to undetectable levels in the liver of HBV-infected mice. In contrast, the average intrahepatic amounts of cccDNA were not reduced compared to the untreated control group. Moreover, immunofluorescence staining of SMC5/6 combined HBV RNA expression analysis on a single-cell-level revealed that this host restriction factor was exclusively detected in hepatocytes without active HBV transcription, thus pointing out the mutual exclusive presence of HBV RNA and SMC5/6. This finding was in line with a previous in vitro study showing that both the use of siRNA targeting the viral HBx mRNA or the host DDB1 mRNA led to an increase in SMC5/6-positive hepatocytes [25]. Thus, these studies show that siRNA-mediated suppression of HBx enabled not only the reduction of viremia and antigenemia but also the prompt reappearance of the SMC5/6 complex in PHH. Chromatin-immunoprecipitation (ChIP-qPCR) assays also confirmed the recruitment of SMC5/6 onto the cccDNA, thus providing further evidence that siRNA therapies targeting mRNA translating HBx can induce silencing of cccDNA transcription in vivo. The fate of SMC5/6 and cccDNA activity was also assessed after treatment withdrawal, either in the presence or absence of the viral entry inhibitor Bulevirtide (BLV, Hepcludex), which is a synthetic myristoylated peptide derived from the pre-S1 domain of the large surface protein of the HBV. BLV specifically binds the cellular receptor (NTCP), thus efficiently blocking the entry of HBV and HDV into the hepatocytes [77]. In 2023, BLV received full marketing authorization in Europe and is recommended to treat CHD in patients with compensated liver disease [78,79]. Notably, the preclinical study of Allweiss, Giersch et al. demonstrated that the cccDNA suppression achieved upon siRNA treatment could be maintained during follow-up for several weeks when new infection events were prevented by administering BLV. However, the study could not assess whether new infections were necessary to reactivate cccDNA molecules that had been silenced through siRNA treatment or whether new incoming HBV DNA-containing virions merely established new transcriptionally active cccDNA molecules. In both cases, the study indicated that shielding the hepatocytes from new infection events plays a crucial role in maintaining transcriptional silencing of the HBV minichromosome. 

While siRNA targeting HBx mRNAs have shown their ability to induce degradation of HBV transcripts (primary mode of action) and to promote SMC5/6-mediated silencing of the cccDNA by abrogating HBx production (secondary mode of action), further studies are needed to explore the direct antiviral effects of ASOs such as Bepirovirsen, as well as their immunomodulatory properties in non-parenchymal liver cells. The induction of toll-like receptors may induce the interferon system, as determined in a study assessing the effectivity of a TLR agonist [80], and thus elicit or further increment direct antiviral effects within the hepatocytes.

### 5.3. Interferon-Based Treatments

Interferons are a class of essential cytokines of the immune system that can induce strong antiviral effects. Human interferons are classified as type-I, type-II, and type-III. While type-I IFNs bind a heterodimeric receptor (IFNAR1 and IFNAR2) expressed on the surface of nearly all cells, the receptor of type-III IFNs (lambda IFN) is expressed primarily on epithelial cells, such as hepatocytes. The binding to their receptors results in the induction of hundreds of so-called interferon-stimulated genes (ISGs). HBV is known to be a weak inducer of ISGs and IFN response. Consequently, the production of endogenous IFNs in infected cells is generally very low, with differences varying depending on the experimental system, infection setting, and patient cohorts (for a review, see also Novotny et al., 2021 [81]). Although the effect of IFNλ on the cccDNA has been less well characterized, both type-I and type-III interferons have been shown to lower HBV replication in vitro and in vivo and to act through a variety of mechanisms, including RNA degradation, transcriptional repression of the cccDNA, association with histone deacetylation, recruitment of transcriptional repressors, and partial degradation of the cccDNA via apolipoprotein B mRNA editing enzyme, catalytic polypeptide (APOBEC) 3A [9,57,82].

Although the wide range of functions of the type-I interferon pegIFNα in HBV infection is still not fully understood [83], at present, the use of pegIFNα remains the only approved therapy for CHB with the advantage of a finite treatment duration and higher rates of HBsAg loss and seroconversion in comparison to NAs [56]. Apart from eliciting immune modulatory effects, as mentioned above, numerous studies demonstrated the ability of pegIFNα to act as an antiviral by suppressing cccDNA activity [7,57,84]. Of note, administration of pegIFNα in HBV-infected human-chimeric mice also promoted the reduction of HBx, the subsequent reappearance of SMC5/6, and the silencing of the cccDNA [55]. However, it remains unclear whether IFNα treatment directly induced the HBx decrease and subsequent SMC5/6 re-appearance or whether the presence of SMC5/6 on the cccDNA was instead the result of distinct transcriptional-repressing events caused by this therapeutic cytokine, which led to the reduction of all HBV RNAs and substantial HBx reduction. Either way, this in vivo study demonstrated that new infection events were central to cccDNA reactivation. 

In clinical trials, pegIFNλ failed to meet non-inferiority criteria in CHB patients compared to pegIFNα [85], and no trials are currently evaluating this therapy for HBV mono-infection. Nevertheless, further evaluation of the mechanisms by which type-III interferons could impact cccDNA activity remains important as it may reveal the potential use of targeting the IFNλ signaling in future combination therapies.

### 5.4. Epigenetic Targeting of the HBV Reservoir

The durability and epigenetic impact of SMC5/6-mediated silencing of the cccDNA still need to be fully understood. However, strategies aiming at promoting an irreversible inactivation of the cccDNA, for example, through the deposition of repressive PTMs, are in principle attractive [86]. Approaches inducing epigenetic suppression of the cccDNA may also act synergistically with SMC5/6, for instance by enhancing the silencing process of SMC5/6. While HBx is suggested to initiate the degradation of SMC5/6 even at low expression levels, epigenetic modifiers would silence the minichromosome by acting independently of HBx expression. Various studies investigated the potential of epigenetic modifiers to suppress HBV replication [87]. These included the inhibition of epigenetic writers like p300/CBP by C646 [7] or erasers like SIRT2 deacetylase by AGK2 [88] or KDM5 by GK5801 [89,90]. Despite encouraging results, drugs that interfere with epigenetic regulators must be seen critically, as they raise substantial safety concerns due to their potential impact on tumor-suppressor genes. Moreover, issues related to drug delivery and target specificity remain challenging [86]. In the last years, alternative promising approaches have emerged in the field of gene editing. Combined with CRISPR technology, site-specific epigenetic editing elicited by fusing epigenetic effector domains to nuclease-deficient Cas9 (dead Cas9/dCas9) results in activated (CRISPRa) or repressed (CRISPRi) transcription [91]. Approaches based on the rewriting of the DNA without inducing cleavage, such as cytosine base editors (CBEs), which convert cytosine into thymine within DNA, may greatly reduce the risk of causing host and virus genome rearrangements [92]. A pioneering study [93] showed the ability of CBEs to reduce HBV expression in cells harboring integrated HBV genomes by targeting the polymerase and surface genes, thus highlighting the potential of these approaches for targeting integrated HBV DNA sequences. More recently, substantial cccDNA editing and reduced viral markers were shown in vitro, in PHH, and even in vivo, using the HBV minicircle mouse model [94]. Considering that in vivo delivery remains challenging for therapeutic applications [95], studies evaluating distinct delivery strategies in models supporting HBV infection and cccDNA generation would be of great value to thoroughly assess the efficacy of CBE-based approaches on infection-derived cccDNA molecules, as well as the stability of edited minichromosomes in human hepatocytes.

## 6. Conclusions

The cure for HBV infection is challenged by the persistence of the HBV reservoir in the liver of chronically infected individuals. The cccDNA minichromosome remains unaffected (NA-based therapies) or is only marginally affected (pegIFNα) by current, approved therapies. Numerous in vitro and in vivo studies have demonstrated that HBx expression is essential for maintaining cccDNA-mediated HBV transcription. Thus, HBx plays a key role in the HBV life cycle from the early phase of infection establishment through the different phases of CHB. Moreover, numerous studies highlighted the involvement of HBx in the progression of HCC, likely by affecting a myriad of cellular pathways, including the interference with interferon and innate signaling pathways [96]. The finding that HBx provokes the continuous degradation of a cellular complex, SMC5/6, crucial in maintaining the structural integrity of chromosomes, further highlights the potential role of this viral regulatory protein in also affecting host cell genetic stability. Thus, elucidating the interaction between HBx and SMC5/6 and the mechanism of cccDNA silencing is essential to develop and design tailored and more efficient therapies that can either directly target the cccDNA, i.e., by augmenting its degradation, or that can trigger its transcriptional repression to obtain a functional and eventually complete cure for CHB. Since just a few copies of cccDNA appear sufficient for viral rebound after treatment cessation, not only immunological control of the infection but also irreversible silencing of the established cccDNA may be required. Patients with chronic or past HBV infection receiving immunosuppressive therapies are indeed at high risk of HBV reactivation, potentially leading to fatal hepatitis flares.

Newly identified small molecules, treatments based on the use of RNAi, and even pegIFNα have shown their ability to interfere with the transcriptional activity of the cccDNA by harnessing the HBx-SMC5/6 axis. While some molecules directly target interactions between HBx and DDB1 or with the cullin4-RING E3 ubiquitin ligase, RNAi revealed its effectiveness by acting upstream, by abrogating HBx production. Further preclinical and clinical studies are, however, needed to explore the effectiveness of these treatments in the liver of CHB patients and to characterize the epigenetic landscape of the cccDNA resulting from these treatments. However, the current need for liver tissue and accurate cccDNA level quantitation represents significant challenges. Coexisting replicative intermediates identical to cccDNA often hamper proper PCR-based cccDNA quantification. More sensitive and standardized assays to monitor cccDNA amounts [97], the number of infected hepatocytes harboring cccDNA, and the development of serological biomarkers enabling the reliable assessment of cccDNA activity in the livers of CHB patients are urgently needed to estimate the effectiveness of therapies in development.

While new emerging therapeutic approaches are encouraging, their safety profiles and the potential for a functional cure require further investigation and exploration of their combined efficacy. A big open question that remains to be answered is the fate of eventually silenced cccDNA. It is unclear whether silenced cccDNA molecules are less stable and, therefore, more prone to degradation and loss through cell division [98] or if silenced cccDNA may be less susceptible to further interventions and/or remain silenced inside the hepatocytes as an inactive cccDNA pool, which can eventually reactivate after years off-treatment or after immunosuppression [99,100]. Moreover, more research is needed to understand the impact of cccDNA silencing on immune recognition and the restoration of adaptive immune responses, which remain central to controlling the infection. Most likely, combinational therapies combining immunomodulatory and antiviral approaches, or those able to act synergistically in suppressing the cccDNA irreversibly, will be key to obtain more durable silencing effects and immune control.

## Figures and Tables

**Figure 1 viruses-16-01667-f001:**
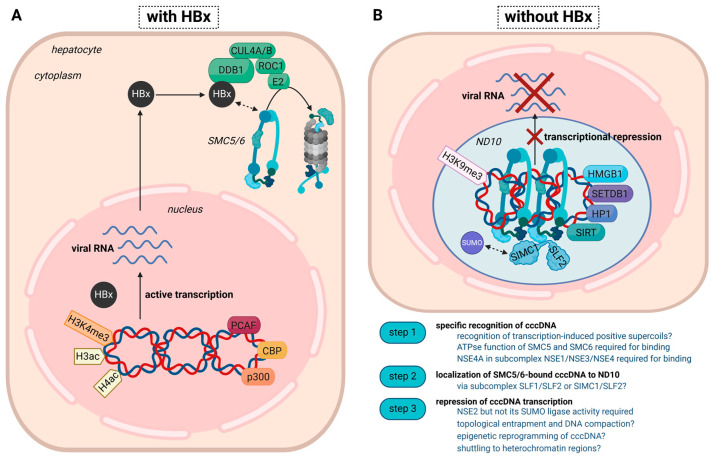
Transcriptional status of the cccDNA minichromosome in the presence or absence of HBx. This proposed model combines established and emerging concepts. (**A**) In the presence of HBx, the cccDNA is actively transcribed and associates with histone-acetyltransferases (PCAF (P300/CBP-associated factor), CBP (CREB-binding-protein), and p300) and active histone marks, including acetylated H3/H4 and H3K4me3. HBx targets SMC5/6 for ubiquitylation by the HBx-DDB1 E3 ligase complex and subsequent degradation by the proteasome. The mechanism of the DDB1-HBx-SMC5/6 interaction awaits further elucidation. (**B**) In the absence of HBx, either achieved through therapeutic intervention or infection with an HBx-deficient virus, the cccDNA is transcriptionally repressed and associates with transcriptional repressors, such as HP1 (heterochromatin protein 1), HMGB1 (High mobility group box 1), SETDB1 (Histone-lysine N-methyltransferase), HDAC1 (Histone Deacetylase 1), as well as the heterochromatin mark H3K9me3. As SMC5/6 is not degraded, it binds the cccDNA and induces transcriptional silencing. The specific recognition of the cccDNA and the orchestration of the subsequent steps, especially the involvement of epigenetic factors/histone PTMs, still need to be fully understood. Whereas the localization of SMC5/6 to ND10 was shown in several studies, the association with SLF2/SIMC1, which might help locate the host restriction factor to highly SUMOylated regions inside ND10, needs to be confirmed in the setting of HBV.

**Figure 2 viruses-16-01667-f002:**
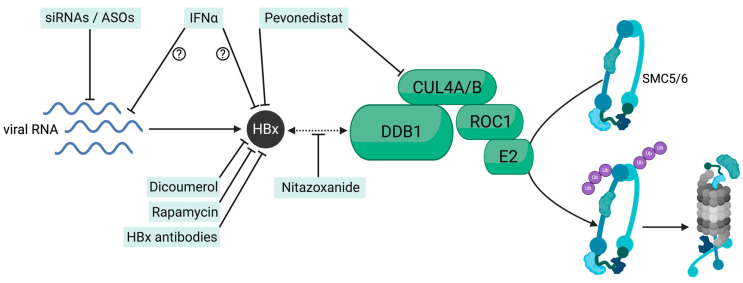
Schematic presentation of therapeutic targets in the DDB1-HBx-SMC5/6-axis. Hindering HBx production with siRNA (small interfering RNA) or ASOs (antisense oligonucleotides), lowering its stability (Dicoumerol, Rapamycin, Pevonedistat), directly inhibiting its function (HBx-specific antibodies) or interfering with the binding of HBx to DDB1 (Nitazoxanide) inhibits HBx-DDB1 E3 ligase complex-mediated degradation of SMC5/6 via the proteasome. The exact mechanism of the anti-HBV activity of interferons is not fully understood but likely involves transcriptional repression or direct HBx inhibition.

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
