# Peer review of "SMC5/6-Mediated Transcriptional Regulation of Hepatitis B Virus and Its Therapeutic Potential"

_viruses, 2024, doi:10.3390/v16111667_

Round 1
Reviewer 1 Report
Comments and Suggestions for Authors
This review provides a comprehensive and insightful overview of the current understanding of the role of SMC5/6 in HBV biology. It effectively illustrates the complex interplay between host factors and viral genome, offering a solid foundation for exploring its therapeutic potential. However, there are some areas that could be improved.
Specific comments
1. The last paragraph in the “Introduction” section is too lengthy. It can be considered to separate the content of current approved therapies from the introduction of SMC5/6.
2. It would be more logical to first introduce “SMC5/6 complex and its functions on cellular chromatin”, followed by discussions on its interaction with HBV in the sections “Mechanism of HBx-mediated degradation of SMC5/6” and “Mechanism of SMC5/6-mediated silencing of cccDNA”.
3. Figure 1:
3.1 In Figure 1(a), HBx, PCAF, CBP and p300 should be shown binding with cccDNA in the nucleus. Similarly, some transcriptional repressors in Figure 1(b) are also.
3.2 The color of SMC5/6 in Figure 1(b) is very similar to that of ND10. May consider to change one of the colors to improve differentiation.
3.3 The text labels in the Figure 1 are too small. Please consider increasing the font size for better readability.
Reviewer 2 Report
Comments and Suggestions for Authors
The authors of the review article "SMC5/6-mediated transcriptional regulation of HBV and its therapeutic potential" have provided an extensive literature review combined with mechanistic analysis of the role of the SMC5/6 complex in HBV lifecycle. This is a very significant topic in the field and the authors have covered multiple aspects of it including the virus and host sides. One suggestion that would make it easier to read for the non-expert or the audience that is not 100% familiar with the details is to better represent the mechanistic details in Figure 1 by adding another panel for example. There is just a lot of information in the text that it makes difficult to follow in these simplified figure and an additional panel with step-by-step references to the text would be helpful.
Comments on the Quality of English LanguageThere are only some minor typos otherwise the manuscript is very well written.
